# Hepatitis Delta Virus–Host Protein Interactions: From Entry to Egress

**DOI:** 10.3390/v15071530

**Published:** 2023-07-11

**Authors:** Susannah Stephenson-Tsoris, T. Jake Liang

**Affiliations:** Liver Diseases Branch, National Institute of Diabetes and Digestive and Kidney Diseases (NIDDK), National Institutes of Health, Bethesda, MD 20892, USA; suzie.stephenson@nih.gov

**Keywords:** hepatitis, viral infection, replication, life cycle, host factors, therapeutic development

## Abstract

Hepatitis delta virus (HDV) is the smallest known human virus and causes the most severe form of human viral hepatitis, yet it is still not fully understood how the virus replicates and how it interacts with many host proteins during replication. This review aims to provide a systematic review of all the host factors currently known to interact with HDV and their mechanistic involvement in all steps of the HDV replication cycle. Finally, we discuss implications for therapeutic development based on our current knowledge of HDV–host protein interactions.

## 1. Introduction

Hepatitis delta virus (HDV), a circular single-stranded RNA virus that uses hepatitis B virus (HBV) co-infection to propagate, causes the most severe form of human viral hepatitis. HDV is also the smallest known human virus, at ~1680 nt long, only producing one protein, hepatitis delta antigen (HDAg), of which there are two isoforms, HDAg-S and HDAg-L. Since HDV produces only one protein, and HDAg appears to be an RNA- and host-factor-binding protein with no known enzymatic function [1], host proteins are required for the HDV replication cycle. In addition, being an RNA virus, HDV promotes the induction of host antiviral proteins. These proteins may interact with the circular RNAs (the genome and antigenome), the mRNA, the HDAg, the ribonucleoprotein (RNP) complex, or both the RNA and HDAg, separately. The host proteins that interact with HDV during its replication are largely unknown. The HDV–host protein interactome has been previously reviewed [2,3,4]. Recent advances in HDV research methods, including genome-wide knockdown screening and mass spectrometry, along with the discovery of multiple non-human deltaviruses [5,6,7,8], have provided new insights into this topic. In this review we summarize the current knowledge on HDV–host protein interactions and discuss future directions of HDV–host protein interaction research. We provide a systematic review of all the known host factors and their mechanistic involvement in various steps of the HDV life cycle (Figure 1) and discuss implications for therapeutic development based on these interactions.

## 2. Review

### 2.1. Viral Entry

HDV virions are thought to mainly enter hepatocytes via the sodium taurocholate co-transporting polypeptide (NTCP) receptor found in the basolateral membrane of hepatocytes, using the preS1 domain of the HBV large surface antigen (HBsAg-L) [9]. However, HDV may not rely solely on HBsAg-L-dependent NTCP-mediated entry, as HBV has been shown to exhibit NTCP-independent entry in extrahepatic tissues, suggesting a similar possibility for HDV [10]. Also, HDV, as well as the more recently discovered other members of the Deltavirus genus, might be able to use viruses other than HBV, or other Hepadnaviruses, as a helper virus, suggesting that HDV may also be able to enter other cell-types via the receptors that are used by the other potential helper viruses [8,11]. Additionally, recent instances of virologic breakthrough of HDV infection during Myrcludex B extended therapy have occurred and appear to be becoming more prevalent. This breakthrough has been shown to be independent of mutations in the HDV genome, suggesting an NTCP-independent pathway for HDV persistence in humans. 

Once inside the cell, HDV RNA is transported to the nucleus, facilitated by its interaction with HDAg. HDAg has a classical type 1 nuclear import signal (NLS) located at amino acids 66–75, which is required for nuclear entry of HDV RNPs using karyopherin (importin) 2α, a protein that is part of the nuclear pore complex and mediates import of cytoplasmic molecules into the nucleus [2,12]. Once inside the nucleus, the RNA is replicated by host RNA polymerase(s).

### 2.2. Viral Transcription

The HDV mRNA, that has an antigenome polarity, is transcribed from the HDV genomic RNA and is 5′ 7 mG capped and 3′ polyadenylated [13,14]. Therefore, the HDV mRNA is transcribed by host RNA polymerase II (RNAPII), as RNAPII is the only host polymerase that interacts with capping enzymes to produce capped RNAs [15]. Other proteins with possible involvement in mRNA transcription include those proteins involved in the initiation, elongation and termination of RNAPII transcription, such as helicases (CCAR1 and CDC5L), hnRNP proteins and transcription factors [16,17,18].

Many host proteins involved in mRNA splicing, such as arginine/serine-rich splicing factor (ASF), heterogeneous nuclear ribonucleoprotein L (hnRNP L), heterogeneous nuclear ribonucleoprotein D (hnRNP D), zinc finger protein 326 (ZNF326), CDC5L, SF3B155, SC35 and ELAV-like protein have been shown to interact with HDV RNA, HDAg or both. Arginine/serine-rich splicing factor (ASF) interacts with the right terminal stem-loop domain of the HDV genomic RNA in vitro, and with both polarities of HDV RNA in HeLa cells by various approaches including mass spectrometry of UV-crosslinked HDV ribonucleoprotein complex, RNA affinity chromatography and screening of a library of RNA binding proteins. The binding was confirmed by RT-PCR of co-immunoprecipitated HDV RNA [17]. In the same study, the hnRNP subunit, hnRNPL, which plays a role in the formation, packaging, processing and function of mRNA, was also found to interact with the right terminal stem-loop domain of the HDV genomic RNA in vitro and with both polarities of HDV RNA in HeLa cells [17]. The hnRNPD and zinc finger protein 326 were identified to have an increase in abundance due to HDV RNA replication in the MALDI-TOF proteomics screen [19] and hnRNPD caused a decrease in genomic RNA accumulation in the combined proteomic–RNAi screen [16]. CDC5L was also detected in the same combined proteomic–RNAi screen performed [10]. SF3B155, a U2 snRNP protein necessary in pre-mRNA splicing, was found to interact with HDV genomic RNA using a yeast three-hybrid system screen of a human liver cDNA library against HDV genomic RNA. This interaction was confirmed in Huh7D12 cells via co-immunoprecipitation of HDV RNA with an anti-SF3B155 antibody and HDV genome-specific PCR [20]. SC35, another host splicing factor, was found to co-localize with HDAg-S by immunofluorescence, only when HDV RNA genome was also present. The co-localization was disrupted when HDAg-L begins to accumulate, along with a decrease in HDV RNA replication, leading the authors to suggest that the speckles in which SC35 is found might be the site of HDV RNA transcription by RNAPII and/or the site of HDV RNA processing [21]. ELAV-like protein 1, an RNA-binding protein that stabilizes AREs-containing mRNAs, was found to be downregulated in the presence of HDV replication in HEK-293 cells via an MS-based quantitative proteomics approach [22]. It is interesting that these splicing associated factors interact with the HDV RNA, HDAg or both, despite no knowledge of the HDAg mRNA (or circles) undergoing splicing.

As the HDV mRNA is transcribed by host transcriptional machinery, it is recognized as a host transcript and able to be translated by host translational machinery. HDV has been found to interact with host translational machinery such as eukaryotic elongation factor 1A1 (eEF1A1), EIF3D, polyadenylate binding protein (PABP) and eukaryotic translation initiation factor 2 subunit 1. While PABP (found to interact with HDV using a combined proteomic–RNAi screen) helps direct the binding of translation factors to the poly-A-tail of mRNAs, eukaryotic translation initiation factor 2 subunit 1 (found to interact with HDV via proteomic screen) and eEF1A1 (found to interact with HDV RNA using mass spectrometry and confirmed with co-immunoprecipitation) aid the initiation and elongation, respectively, by delivering tRNAs to the 40S ribosomal subunit [16,17,19]. EIF3D, found by Mendes et al., 2013 [22] to interact with HDV through mass spectrometry, is a translation initiation factor that binds to the 40S ribosomal subunit. Translation of the mRNA produced from unedited genomes early in infection leads to production of the HDAg-S.

### 2.3. HDAg-S Functions

Recently, Janus Kinase 1 (JAK1) was shown to be functionally involved in HDAg-S mediated HDV replication (Figure 2). This interaction was linked to the phosphorylation of extracellular-signal-related kinases 1 and 2 (ERK1/2) by JAK1 [23]. ERK1/2 have been previously shown to interact and phosphorylate a serine at position 177 (S177) of HDAg-S via co-immunoprecipitation analysis. Phosphorylation was confirmed with an in vitro kinase assay, in which Flag-ERK1 or Flag-ERK2 directly phosphorylated S177 of HDAg-S, and this phosphorylation was detected with immunoblotting using an antibody against phosphorylated S177 HDAg-S and mass spectrometry. Additionally, they found that phosphorylation of S177 of HDAg-S increased genomic but not antigenomic RNA replication. Finally, they found that phosphorylation of HDAg at S177 leads to the interaction of the HDAg-RNA RNP with RNAPII, thereby aiding in HDV replication [24] (Figure 2).

Other host proteins have been found to post-translationally modify HDAg-S, including Casein Kinase II (CKII), PKR, PKC, PRMT1, SUMO1 and Ubc9 [2]. More specifically, CKII, PKR and PKC phosphorylate HDAg at S2 and S213, S177, S180 and T182, and S210 respectively. Phosphorylation at these sites by CKII and PKC was found to positively modulate HDV RNA replication, while phosphorylation by PKR leads to decreased RNA replication as suppression of HDAg phosphorylation by PKR leads to an increase in HDV replication [25,26]. Protein arginine methyltransferase 1 (PRMT1) methylates host proteins at arginine residues and methylates HDAg at R13, regulating subcellular localization and RNA replication, as a mutation in R13 inhibits HDV RNA replication and prevents the ability of HDAg to form a speckled structure in the nucleus [27]. Finally, the sumoylation proteins, Small ubiquitin-related modifier isoform 1 (SUMO1) and Ubc9, sumoylate multiple lysine residues of HDAg, increasing HDV RNA synthesis, specifically the genome and mRNA [28].

HDAg has also been shown to interact with various proteins involved in chromatin remodelling, including the histones H1e, H2A and H4, and the histone-interacting proteins Ying-Yang 1 (YY1), CRB, p300, histone H1 binding protein (NASP), nucleolin (C23), nucleophosmin (B23), chromodomain helicase-DNA-binding protein 4 (CHD4) and BAZ2B. While these proteins interact with HDAg directly, they likely have a role in HDV RNA replication, similar to their role in histone modification and chromatin remodelling. HDAg-S was found to bind to histone H1e at the N-terminal 67 amino acids using tandem affinity purification and mass spectrometry; conversely, such an interaction barely occurs with HDAg-L. This binding was confirmed by mutation experiments, in which HDV replication was inhibited by N- or C-terminal deletion mutants of histone H1e, and was rescued by addition of wild-type histone H1e [29]. Ying-Yang 1 (YY1) co-sediments with both HDAg-S and HDAg-L, along with YY1’s associated acetyltransferases, CREB-binding protein (CRB) and p300. Additionally, overexpression of YY1, CRB or p300 resulted in increased HDV RNA replication [30]. Nucleolin (C23) was first determined to interact with HDAg-S by immunofluorescence staining studies and later confirmed by co-immunoprecipitation studies. The N-terminal domain of HDAg was required for its nucleolin binding. This binding was found to be determined by two nucleolin binding sites on the HDAg—NBS1 and NBS2—both having a conserved core sequence of K(K/R)XK. Both sequences were shown to be required for the replication of HDV RNA [31]. Nucleophosmin (B23) was also found to interact with HDAg, being upregulated in the presence of both HDAg-S and co-sedimenting with both HDAgs. In addition, overexpression of B23, but not a mutated, defective B23, lead to enhanced HDV replication. Altogether, B23 appears to bind to HDAg and upregulate HDV RNA replication; its expression is also upregulated by HDAg [32]. More recently, Abeywickrama-Samarakoon et al., 2020 [33] showed that bromodomain adjacent to zinc finger domain 2B (BAZ2B) protein, a regulatory subunit of BAZ2B-associated remodelling factor (BRF) ISWI chromatin remodelling complexes, binds to HDAg-S, and that shRNA-mediated knockdown or inactivation with an inhibitor of BAZ2B leads to impaired HDV replication. Additionally, they found that the short linear interacting motif (SLiM), KacXXR, of HDAg-S is required for its interaction with BAZ2B BRD and for HDV replication [33]. 

### 2.4. Viral Replication

Host RNAPII is co-opted by the circular HDV antigenome, which is used as a template for transcription of the linear, multimeric HDV genome using rolling-circle replication. RNAPII was first suggested as the host RNA polymerase that replicates the genome after α-amanitin, a known RNAPII inhibitor, was found to decrease the accumulation of genomic HDV RNA (and mRNA) in both cultured cells and nuclear extracts. Later, studies using co-immunoprecipitation, binding assays, mutagenesis and in vitro transcription experiments showed that RNAPII interacted with the terminal stem-loop domains of both polarities of HDV RNA [18,34,35]. The HDV genome is 5′ 7 mG capped, further adding to the likelihood that it is transcribed by host RNAPII [36]. Several proteins have been found to interact with HDV that are important for host RNA transcription, including helicases (CCAR1 and CDC5L), hnRNP proteins and transcription factors [16]. Abrahem and Pelchat 2008 [18] showed that an active RNAPII preinitiation complex containing other transcription factors TFIIA, TFIIB, TFIID, TFIIE, TFIIF, TFIIH and TFIIS binds to a specific region of the RNA genome, which is thought to serve as a promoter. 

In addition, the work of Yamaguchi et al., 2001 [37] suggested that HDAg binds to RNAPII, in direct competition with NELF but not DSIF (Figure 2). In host RNAPII RNA transcription from a DNA template, NELF and DSIF forms a complex and bind to RNAPII, thereby inducing transcriptional pausing. Yamaguchi et al., 2001 [37] suggest that the binding of HDAg directly to RNAPII reverses the pausing effects of DSIF and NELF by displacing NELF, allowing transcription elongation. More recently, the cyclin-dependent kinases, CDK8 and CDK19, which are important for the mediator complex’s ability to release RNAPII from transcriptional pausing, have been shown to be necessary for HDV replication. Inhibition or knockout of both CDKs results in abrogation of HDV replication [38] (Figure 2). Overall, it is clear that RNAPII, its transcription factors, and the many positive and negative regulatory proteins are important in HDV RNA replication.

There is much debate over which host RNA polymerase replicates the HDV antigenome from the genomic template. RNAPII has been found to interact with the genome, antigenome and mRNA (discussed above). Alpha-amanitin studies also showed a decrease in antigenomic RNA, supporting the role of RNAPII [39]. However, the antigenome RNA has not been found to be 5′ capped yet, thus raising question about the involvement of RNAPII. RNAPI and RNAPIII have also been shown to interact with both polarities of the HDV RNA in immunoprecipitation assays [40]. RNAPI has been proposed to be the main transcriber of the antigenome RNA due to the localization of newly synthesized HDV antigenome RNA, metabolically labelled with bromouridine and visualized with immunofluorescence, to the nucleolus, where RNAPI is also localized [41]. Additionally, the authors suggested that, since the RNAPI-associated transcription factor, SL1, was co-precipitated with HDAg, RNAPI must play a role in HDV synthesis [41]. Finally, other studies have shown that transcription of the antigenomic RNA is not alpha-amanitin sensitive, despite previous studies showing the opposite [42]. However, during a combined proteomic–RNAi screen performed by Cao et al., 2009 [16], no RNAPI- or RNAPIII-specific subunits were found to interact with replicating HDV. Overall, it is still unknown which host RNA polymerase synthesizes the HDV antigenomic RNA.

More recently, Verrier et al., 2020 [43] performed a combined small molecule and loss-of-function screen, uncovering an enzyme in the pyrimidine biosynthesis pathway, aspartate transcarbamylase and dihydroorotase (CAD), as important for HDV RNA replication in an uracil-depletion-dependent manner. In addition, they found that estrogen receptor alpha (ESR1), an activator of CAD, was also important for HDV RNA replication. They performed inhibition studies with inhibitors or CAD and ESR1, and found specific inhibition of HDV replication in a dose-dependent manner. Both the genome and antigenome were affected by CAD inhibition, which is logical due to the presence of pyrimidines in both RNAs.

HDV RNA has been shown to interact with the RNA binding proteins, NONO (previously known as P54nrb), SFPQ (previously known as PSF) and PSPC1, and the lncRNA, NEAT1, which are present in nuclear paraspeckles [17,44,45]. Paraspeckles are small, subnuclear bodies found within the interchromatin space, numbering between 5 and 20 foci per nucleus, and contain both RNA and proteins [46]. It is believed that paraspeckles function in transcriptional regulation, mainly by sequestering mRNA, and adenosine–inosine editing of RNAs [46]. Beeharry et al., 2018 [45] demonstrated that SFPQ, NONO and PSPC1 have a role in HDV replication, as knockdown of these three proteins with RNAi inhibits HDV replication. They suggested that HDV replication causes cellular stress, leading to the delocalization of PSPC1 into extra-nuclear large foci, or “stress granules”, in which it co-localizes with PABP (discussed above), causing cell cycle arrest and loss of cellular adherence in HEK 293 cells. They also found that HDV replication causes an upregulation of NEAT1 transcripts, leading to an enlargement of NEAT1 foci, which was consistent with previous reports that NEAT1 levels increase as part of the cellular response to stress events, such as viral infection. Studies with other viruses suggested that sequestration of SFPQ from host genes by NEAT1 could activate the antiviral gene, IL8. Beeharry et al., 2018 [45] also showed that HDV replication increased IL8 mRNA levels approximately two-fold, suggesting that the increase in NEAT1 caused by HDV replication causes SFPQ sequestration and leads to IL8 induction. Overall, these data showed that host paraspeckle proteins and related RNA may play a prominent role in HDV replication.

Finally, during the rolling-circle replication of both the genome and the antigenome, a multimeric linear RNA is produced that is self-cleaved by the genomic and antigenomic ribozymes, leaving a 5′ hydroxyl and a 2′3′ cyclic phosphate at the 3′ end of the linear monomer. A host ligase is required for ligating the 5′ and 3′ end of the linear monomer, but no host ligase has been identified so far. Possible host ligases include DNA ligase 1, which has been shown to act as an RNA ligase in viroids, or C12orf29, the only known human enzyme catalysing ATP-dependent RNA ligation between a 5′ phosphate and a 3′ hydroxyl [47,48].

### 2.5. HDAg-L Function and HDV Assembly

HDAg-L mRNA is created when the antigenomic RNA circle is edited by the double-stranded RNA specific editing enzyme Adenosine Deaminase RNA specific 1 (ADAR1) at the Amber-W site [49,50,51,52]. The editing of this site from an adenosine to an inosine causes the RNAPII to read the base as a guanosine, therefore adding a cytosine to the elongating genome RNA. Then, when the mRNA is translated using the genome as a template, a guanosine is added instead of adenosine, thereby changing the stop codon into a tryptophan and allowing the translation of the additional 18 amino acids [53].

In order to export HDV RNA out of the nucleus and into the cytoplasm, the HDAg-L, bound to the RNA circle as an RNP, interacts with the nuclear export signal-interacting protein (NESI) with its nuclear export signal (NES), which spans residues 198–210 [54]. This chromosome region maintenance 1 (CRM1)-independent NES also allows binding of the cellular export receptor, TAP (also known as NXF1) and its adaptor protein, ALY (also known as REF), which are responsible for host RNA export [55].

Within the additional 19aa of HDAg-L is also an isoprenylation sequence, CXXX, used by host farnesyl transferase [56]. The isoprenylation of the C-terminal domain of HDAg-L allows the interaction of HDAg-L with cell lipid membrane and HBsAg for packaging of HDV RNPs into virions [57,58]. It is not clear how the HDAg-L complexed HDV genome assembles specifically with HBsAg. Whether there is a sequence-specific interaction between HDAg-L and HBsAg is not known. Previous work has shown that the HDV genome RNP can be packaged with the small, medium and large HBsAg, but that only virions with all three envelope proteins showed evidence of infectivity in HDV-permissive cells [59]. Furthermore, the isoprenylation of HDAg-L also inhibits HDV RNA replication [60,61]. 

### 2.6. HDV Cellular Egress

It is not fully understood how HDV exits the cell and this is confounded by the controversial evidence that HBV envelope proteins are not necessarily required for HDV cell egress. In recent studies, HDV has been shown to be able to use non-HBV enveloped viruses, such as flaviviruses, hepaciviruses and vesiculoviruses, to form virions and exit the cell [11]. As all these viruses have their own methods of cellular egress, it is difficult to say, precisely, which method is used in the case of HDV. Hepatitis C virus, in particular, was shown to act as a helper virus for HDV in the liver of co-infected humanized mice. However, these studies are considered controversial and they have not yet been validated by others. 

Even if one focused just on cellular egress of HDV with the use of HBV as the helper virus, there is still more to be understood. Attempting to draw conclusions about HDV cellular egress from HBV cellular egress is not ideal, as research into HBV cellular egress is still ongoing and inconclusive. A recent paper suggests the involvement of the host gene tumour susceptibility gene 101 (TSG101) in HBV release, as knockdown of TSG101 in cells and transgenic mice suppressed HBV release. In addition, ubiquitination of TSG101 and Lys-96 in HBc by UbcH6 and NEDD4, respectively, was required for HBV egress [62]. In addition, it is still possible that the HDV virion does not follow the same pathway to cellular egress that HBV does, as the RNA, HDAg or RNP may interact with different host proteins. As an example, within the additional amino acids of HDAg-L, there is also a clathrin adaptor homology region, which allows HDAg-L to interact with the clathrin heavy chain, possibly facilitating clathrin-mediated exocytosis [63,64].

### 2.7. Host Antiviral Activity

HDV RNA has been shown to be recognized by the double-stranded binding protein, MDA5. Recognition of HDV by MDA5 leads to the induction of interferons beta and lambda [65]. As stated above, PKR phosphorylation of S177 of the S-HDAg leads to suppression of viral replication, as a dominant negative PKR that was unable to phosphorylate S-HDAg leads to an increase in HDV RNA replication [26]. A detailed discussion of the anti-HDV functions and mechanisms of interferon stimulated genes is beyond the scope of this review. We would refer the reviewers to some recent publications on this topic [66,67].

## 3. Conclusions and Future Directions

While many proteins have been found to interact with either the HDV RNA, HDAg or both, a fair amount of these proteins have been found with screens; therefore, it is unknown exactly how they interact with HDV during its replication cycle as many have not been confirmed with additional experimentation. In addition, many of the studies performed do not differentiate between the HDV mRNA, antigenome and genome.

Despite recent advances about these proteins, there are still many aspects of the HDV replication cycle that remain a mystery, including the ligase that ligates the linear monomeric genome and antigenome, which RNA polymerase truly transcribes the HDV antigenomic RNA and how the host RNAPII, a DNA-dependent RNA polymerase, uses the HDV RNA circle(s) to replicate HDV.

Knowing which host proteins interact with HDV during its replication cycle is important, not only for understanding the replication cycle but also as drug targets since HDV, by itself, has very few targetable proteins. Two available drugs, lonafarnib and Myrcludex B, do just that, targeting the host proteins farnesyltransferase and NTCP, respectively. However, while these drugs are able to prevent viral egress and entry, respectively, there are currently no licensed drugs that affect the in-cell replication cycle of HDV. In addition, as discussed before, the breakthrough of HDV infection during extended Myrcludex B treatment suggests that monotherapy may not be sufficient. Therefore, understanding which host proteins interact during the entire HDV replication cycle is of the utmost importance for treating HDV infections, even with drugs currently available.

## Figures and Tables

**Figure 1 viruses-15-01530-f001:**
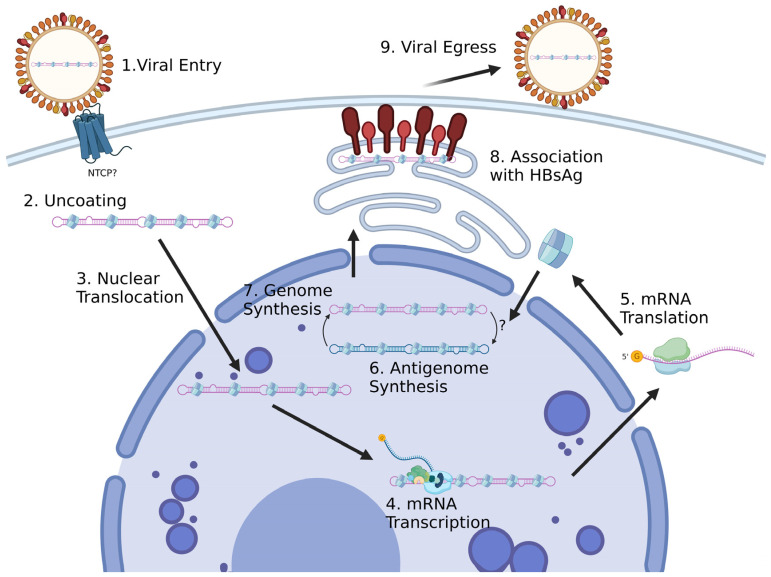
Schematic of the replication cycle of hepatitis delta virus. Created with Biorender.com.

**Figure 2 viruses-15-01530-f002:**
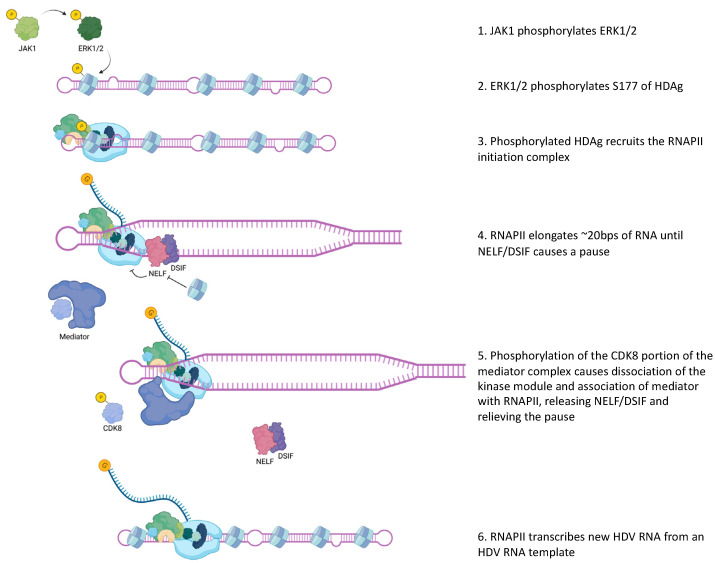
Proposed mechanism of RNAPII initiation on HDV RNA template. Created with Biorender.com.

## Data Availability

Not applicable.

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
