# Peer review of "Hepatitis Delta Virus–Host Protein Interactions: From Entry to Egress"

_viruses, 2023, doi:10.3390/v15071530_

Round 1

Reviewer 1 Report

The authors have assembled a well laid out and detailed review of HDV / host protein interactions.  The following comments outline minor modifications required:

1   The authors should be careful in suggesting that HDV entry occurs only via the NTCP receptor.  While NTCP is clearly a high affinity receptor for L-HBsAg dependent entry of HBV and HDV, there are well documented examples of NTCP-independent entry of HBV in extrahepatic tissues (PMID 29785877) which suggest a similar possibility for HDV.  The ability of HDV to propagate via HBsAg independent means is also well known (PMID 31068585).  Additionally, L-HBsAg is not required for the formation of HDV (PMID 8416375) and L-HBsAg deficient HDV which is self sustaining in the presence of bulevirtide has been described.  Moreover, real virologic breakthrough of HDV infection to bulevirtide with extended therapy is now becoming much more prevalent and this has recently been shown not be driven by mutational alterations in the HDV genome – these clearly identify a NTCP-independent pathway for HDV persistent in humans.

2.     The authors should include a small section of HDV envelopment and secretion, this is the only part of the HDV life cycle which is missing.

Reviewer 2 Report

This  review describes in detail the current information  on cellular proteins involved in the replication of HDV RNA and  their interaction  with the single protein of HDV during the process of viral maturation.

This knowledge on interaction of  host proteins  with HDV is very important to  improve drugs to treat chronic HDV infection. 

Author Response

Thank you for taking the time to review our paper and for your favourable comments suggesting that this review is both detailed and important knowledge. We have added significantly to the sections on viral entry, HDAg-L function and HDV assembly. In addition, we have added a new section, HDV Cellular Egress. We also included some additional references where necessary. We believe that, overall, these additions enhance the organisation and comprehensiveness of the review, as well as the adequacy of the references.